# Analysis and Verification of Load–Deformation Response for Rocking Self-Centering Bridge Piers

**Shijie Wang [1], Zhiguo Sun [1,\*] and Dongsheng Wang [2]**

[1]  Key Laboratory of Building Collapse Mechanism and Disaster Prevention, Institute of Disaster Prevention, China Earthquake Administration, Beijing 065201, China

[2]  School of Civil and Transportation Engineering, Hebei University of Technology, Tianjin 300401, China

\*  Correspondence: sunzhiguo@cidp.edu.cn; Tel.: +86-189-4113-4800

**Abstract:** Rocking self-centering (RSC) bridge piers were proposed based on the bridge seismic resilience design theory, pushing the development of bridge sustainability. To develop a seismic design method for RSC bridge piers, a clear understanding of their behavior under earthquakes is essential. This study analyzed the whole lateral force–displacement response of RSC piers, taking into account both rotational and flexural deformation, which resulted in a clearer understanding of their behavior under seismic actions. In this study, the whole loading process was simplified into three statuses, and a calculation method was developed to determine the relationship between lateral force and displacement of both single-column and double-column RSC bridge piers. The accuracy of the proposed method was verified by comparing the calculated results with experimental data for six single-column and two double-column RSC bridge piers. The results show that the proposed calculation method predicts the initial stiffness, yield and peak loads, and yield and peak displacements well for RSC bridge piers. The method offers valuable insights into the seismic response of RSC bridge piers, which can serve as a reference for future research, promoting the safety and stability of these structures.

**Keywords:** calculation method; force–displacement response; mechanical behavior; rocking self-centering bridge piers; safety and stability; seismic behavior

## 1. Introduction

Bridges are an essential part of transportation infrastructure, and their seismic performance is critical for ensuring the safety of transportation systems. Traditional reinforced concrete bridge piers, designed using ductile seismic design theory, consume seismic energy through plastic hinge deformation during strong earthquakes, which can cause significant damage and result in long periods of post-earthquake repair [1–3]. Material deterioration and post-earthquake damage pose challenges to the sustainability of structure functions. In contrast, rocking self-centering (RSC) piers are a novel pier structure based on bridge seismic resilience design theory. RSC piers dissipate energy and provide seismic isolation by oscillating between the pier and foundation during strong earthquakes. Unbonded prestressing tendons (PTs) ensure self-centering capability, while energy dissipation (ED) steel bars and other devices provide energy dissipation capacity. The RSC pier effectively reduces damage to concrete and steel, significantly shortening the time required for bridge repair after an earthquake [4], and the piers are assembled on-site, significantly reducing construction time and minimizing environmental impact, aligning with the low-carbon and environmentally friendly cause. RCS structures offer an excellent solution to enhance the ability of bridges to withstand natural disasters such as earthquakes and to develop reliable, sustainable, and high-quality structures.

Researchers have utilized numerical analysis methods, such as the centralized plastic hinge method [5], fiber element method [6], and solid element method [7], to study the

seismic behavior of RSC piers [8]. However, these methods require advanced mathematical and programming skills and significant investment of time and effort, making them difficult for engineers to use. The existing experimental research on RSC piers has primarily focused on quasi-static testing of single-column piers and lacks reliable studies on the behavior of the entire bridge system under dynamic loads. Furthermore, it should be noted that the design and calculation methods for RSC bridge piers are still in their early stages, and further research and exploration is necessary. One potential avenue for progress in this area may be the use of integrated machine learning and optimization approaches [9–12].

RSC piers provide a promising solution to reduce damage to bridges during strong earthquakes, and an easily applicable load–deformation analysis method is crucial for their practical application. Various methods have previously been derived, such as calculation of node rotation and overturn analysis of precast segmentally assembled RSC piers proposed by Bu et al. [13], and a calculation method for new base rocking isolation piers proposed by Guo [14]. Wang et al. [15] proposed a target displacement calculation method for RSC piers based on the structural damage level, assuming the formation of a plastic hinge at the bottom of the pier using an equivalent plastic hinge model. However, due to the opening–closing behavior of RSC piers during an earthquake, the formation of a plastic hinge at the bottom of the pier may be difficult. Han et al. [16] proposed a calculation method for the three statuses of the RSC piers based on their mechanical behavior. This method uses the empirical compression zone height for the calculation, which may lead to significant deviations when calculating different types of RSC piers. Moreover, previous research on calculation and analysis has been mainly based on theoretical methods and lacks sufficient experimental validation.

This study aimed to ameliorate the disadvantages of previous studies on RSC piers. In addition to analyzing the RSC single-column pier, this research also examines double-column RSC piers and verifies the analysis results using experimental data. Although various calculation methods have been proposed in earlier studies, there is a need for further improvement to ensure their accuracy and applicability to different types of RSC piers. This research focused on developing more effective calculation methods that can accurately assess the whole load–deformation response of RSC piers to ensure their reliable and safe response during strong earthquakes.

## 2. Analysis of the Lateral Force–Displacement Response of RSC Bridge Piers

### 2.1. Analysis Method of the Force Distribution and Behavior of RSC Bridge Piers

There are currently three main theoretical methods for calculating the force–displacement relationship of RSC bridge piers. The first method uses rigid body rotation theory [17], in which the compression is assumed to be zero. This method has a simple model and calculation procedure, but significantly overestimates the initial stiffness and strength of the RSC bridge piers. The second method uses the monolithic beam analogy [18], in which the compression zone height is determined by iteration [19] by moment balance at the rocking interface. This method provides more accurate results, but involves iteration and selection of material constitutive relations in its calculation procedure. The third method falls between the first two methods and calculates the force–displacement curve using an empirical formula for determining the compression zone height [7]. However, it is not suitable for RSC bridge piers with different structural forms. Essentially, all the three methods calculate the pier displacement after determining the compression zone height of the joint section. This paper is based on the second method and considers both the mechanical behavior of RSC bridge piers and the damage to structural materials. The deformation of RSC bridge piers is simplified into two parts: rotational deformation caused by rocking response at the bottom (or top) of the pier, and the flexural deformation of the piers. This simplification allows for a force–displacement relationship of RSC bridge piers to be calculated in three stages that are more usable to engineering application.

The main differences between RSC bridge piers and conventional reinforced concrete bridge piers lies in the fact that the former enables rocking response by relaxing the

connection between the pier base and the foundation. Moreover, unbonded prestressing tendon piers provide self-centering capability, while the reinforcement or ED devices provide energy dissipation capacity. As a result, the RSC bridge pier and conventional reinforced concrete bridge piers exhibit different seismic responses.

The unbonded ED bars in the joint section lead to uncoordinated strains between the concrete and ED bars, and uncoordinated strains on both sides of the neutral axis. Hence, the assumption of a plane section remaining plane in traditional reinforced concrete structural section analysis is not applicable, and conventional bending moment–curvature analysis cannot be used. Consequently, the top displacement of RSC bridge piers can be regarded as a combination of the rotational displacement caused by the rotation angle at the joint section and the flexural deformation of the piers, as illustrated in Figure 1.

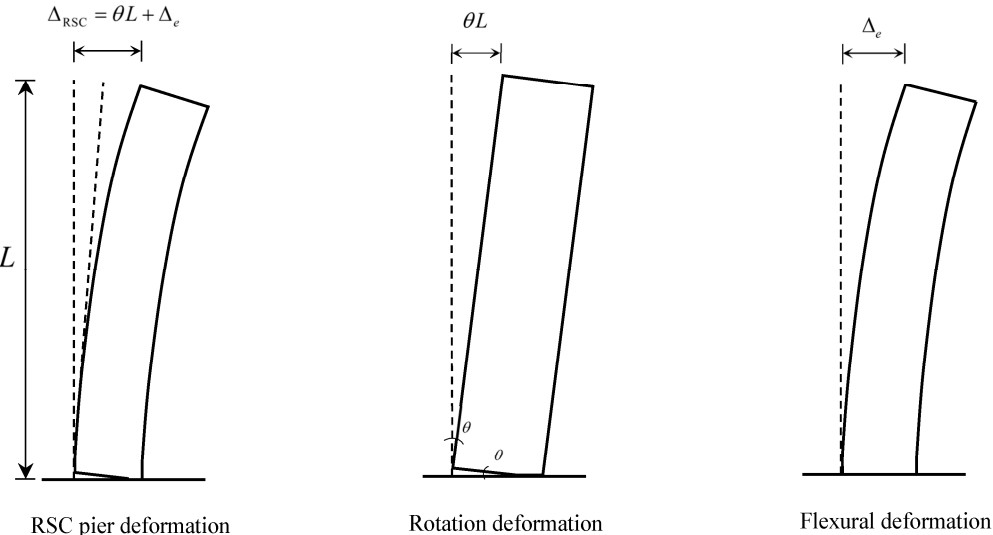

**Figure 1.** Deformation of RSC bridge piers.

This article presents a three-stage analysis method to calculate the lateral force–displacement relationship of an RSC rectangular bridge pier with embedded ED bars. It should be noted that the analysis method for an RSC bridge pier with other ED components is similar to that for a pier with ED bars and only requires changing the stress–strain relationship of the ED bars to that of other ED components.

In the case of ED bars under compression at the joint, their combined force can be neglected in the calculation due to their low reinforcement ratio and the presence of the unbonded section, which further reduces the strain of the compression steel bars [20–22]. For anti-buckling ED components, the combined force of compression-side ED components cannot be ignored. The strain of the compression ED component can be calculated based on the curvature at the bottom of the pier, and then the combined force of the compression ED component can be obtained. The experimental verification in this work tested three different forms of ED components in addition to the ED bars, ensuring the applicability of the theoretical method.

### 2.2. Calculation Procedure for the Top Force and Displacement of RSC Bridge Pier

#### 2.2.1. Decompression Status

In the initial status before applying a lateral load, the bridge pier is subjected to the initial tension of the PTs and the weight of the upper structure, as shown in Figure 2. At this stage, the initial strain $\varepsilon_{c0}$ at the bottom section of the pier can be calculated as:

$$\varepsilon_{c0} = \frac{W + F_{PT0}}{E_s A_s + E_c A_c + E_{PT} A_{PT}} \tag{1}$$

$$A = A_c + A_s + A_{PT} \tag{2}$$

where $W$ represents the weight of the upper structure, $F_{PT0}$ represents the initial tension force of the PTs, $E_s$, $E_c$, and $E_{PT}$ represent the elastic modulus of the ED bar, concrete, and PT, respectively, and $A$, $A_c$, $A_s$, and $A_{PT}$ represent the sectional area of the bottom joint of the pier, the net area of concrete, the area of ED bars, and the area of PTs, respectively.

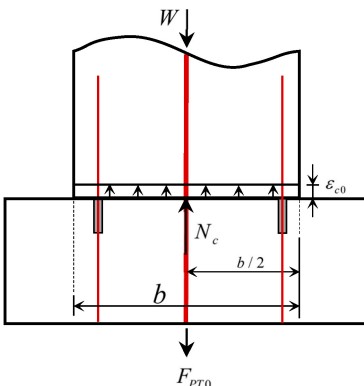

**Figure 2.** Bridge pier base in initial status.

As the loading progresses from the initial status to the compressive strain on the outermost fiber of the bottom section decreased to zero, the bottom section of the pier changes to decompression status, as illustrated in Figure 3. Prior to this state, the entire bottom section of the pier is subjected to compression, and the strain distribution of the bottom section follows the assumption of the plane section remaining plane.

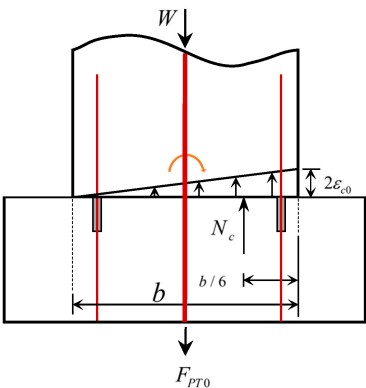

**Figure 3.** Bridge pier base in decompression status.

The curvature $\varphi_d$ and bending moment $M_d$ of the pier bottom section in the decompressed status can be calculated as:

$$\varphi_d = \frac{2\varepsilon_{c0}}{b} \tag{3}$$

$$M_d = \varphi_d EI = \varphi_d (E_s I_S + E_c I_c + E_{PT} I_{PT}) \tag{4}$$

where $b$ represents the section length of the pier in the direction of calculation, $I_s$, $I_c$, and $I_{PT}$ represent the moment of inertia of the all ED bars, the net concrete, and the PTs, respectively, and $EI$ represents the flexural stiffness of the section at the bottom joint of the pier.

In this status, the top displacement of pier is only attributed to the flexural deformation of the pier.

$$\Delta_d = \frac{\varphi_d L^2}{3} \tag{5}$$

The lateral force at the top of the pier is:

$$F_d = \frac{M_d}{L} = \frac{\varphi_d(E_s I_S + E_c I_c + E_{PT} I_{PT})}{L} \tag{6}$$

where $L$ represents the height from the point of the lateral force to the bottom of the pier.

As a result, the lateral stiffness $k_d$ of the pier prior to the decompression status is calculated as:

$$k_d = \frac{F_d}{\Delta_d} = \frac{3EI}{L^3} \tag{7}$$

2.2.2. Yield Status

After the decompression status, the force at the top of the pier increases slightly, due to the concrete is unable to provide tensile stress, causing the bottom interface to start opening up. This opening up of the bottom interface can lead to a reduction in the cross-sectional area of the pier, which can cause a change in its structural behavior. The contact area between the pier and the bearing decreases from full section contact to partial section contact.

When the neutral axis exceeds the position of the ED bars, tension stress occurs in the ED bars. As the lateral force increases, the neutral axis continues to move until the ED bars yield, indicating that the pier has entered the yield status, as illustrated in Figure 4.

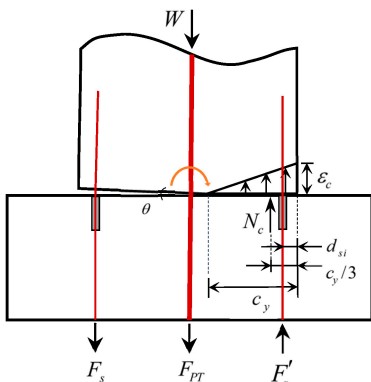

**Figure 4.** Bridge pier base in yield status.

The yield status of the RCS bridge pier can be determined by the yield of the ED bars or other ED components under tension. In this status, the static equilibrium condition of the load can be expressed as:

$$N_c = W + \sum F_{PT} + \sum F_s - \sum F_s' \tag{8}$$

where $N_c$ represents the force of concrete in the pressure zone, $F_{PT}$ represents the combined force of the tensile PTs, $F_s$ represents the combined force of the tensile ED bars, and $F_s'$ represents the combined force of the compression ED bars.

The compression zone height at the bottom of the pier can be calculated as follows:

$$c_y = \frac{N_c}{0.5 f_c h} \tag{9}$$

where $f_c$ is the compressive strength of concrete, $h$ is the section length of the pier in the direction perpendicular to the calculation direction.

When calculating the tensile strain of the ED bar, the strain penetration effect should be considered. The ED bar is deeply embedded in the concrete, as depicted in Figure 5a. In the unbounded segment, the tensile stress of the ED bar is uniform. In the bonding segment, during the process of stress transfer, the strains of the steel and the concrete gradually change, the tensile stress of the steel decreases gradually from the outside to the

inside along the length of the steel, the tension is gradually transferred to the concrete, and the bonding force of the concrete increases overall until the tensile stress of a certain point in the steel is zero, as shown in Figure 5b. Hence, the actual strain penetration length of the steel in the concrete can be equivalent to a length of unbounded segment, denoted as $L_{eu}$ [23] in Figure 5c. When calculating $L_{eu}$, it is important to note that $\varepsilon_s$ changes with $L_{eu}$ and $L_{eu}$ should converge before the calculation.

$$L_{eu} = 0.4 \frac{f_s \varepsilon_s \sqrt{d_s}}{\sqrt{f_c}} \tag{10}$$

$$\varepsilon_s = \frac{f_s}{L_s + 2L_{eu}} \tag{11}$$

where $f_s$ is the stress of the ED bar, $\varepsilon_s$ is the strain of the ED bar, $d_s$ is the diameter of the ED bar, and $f_c$ is the compressive strength of concrete.

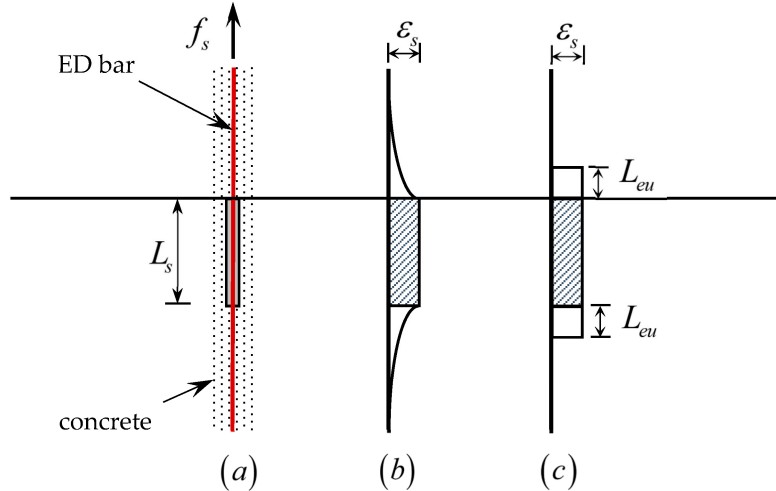

**Figure 5.** Equivalent unbonded length of the ED bar due to strain penetration. (**a**) ED bar embedded in concrete at the joints. (**b**) Variation of strain in ED bars with depth. (**c**) Equivalent unbonded length.

The opening angle at the bottom of the pier can be calculated as:

$$\theta_y = \frac{\varepsilon_s (L_s + 2L_{eu})}{(d_{si} - c_y)} \tag{12}$$

where $Ls$ is the unbonded length of the ED bar, and $d_{si}$ is the distance from the $i$-th ED bar to the compressed zone edge of the pier section.

The yielding behavior of the ED steel bars is modeled using a three-segment model [24], while a two-segment model is adopted for unbonded PT due to the absence of an obvious yielding range. The force of the PTs can be calculated as:

$$\sum F_{PT} = \sum f_{PTi} A_{PTi} = \sum F_{PT0} + \sum E_{PT} A_{PTi} \frac{\theta_y (d_{PTi} - c_y)}{L_{PT}} \tag{13}$$

The force of the ED bar on the tensile side is:

$$\sum F_s = \sum f_{yi} A_{si} + \sum f_{si} A_{si} = \sum f_{yi} A_{si} + \sum E_s A_{si} \frac{\theta_y (d_{si} - c_y)}{L_s} \tag{14}$$

where $\Sigma f_{yi} A_{si}$ represents the yielding force of the $i$-th ED bars on the outer side of the section at the joint, $\Sigma f_{si} A_{si}$ represents the force of the $i$-th ED bars that have not yielded on the tensile side, $\Sigma f_{PTi} A_{PTi}$ represents the force of the $i$-th unbonded PTs, $d_{PTi}$ and $d_{si}$ is the distance from the $i$-th PT and $i$-th ED bar to the compressed zone edge of the pier section.

After that, the compressed zone height $c$ and the bottom angle $\theta$ are calculated iteratively using the aforementioned procedure until the compressed zone height converges. Usually, two to five iterations would be sufficient to achieve desirable accuracy. The curvature at the bottom section of the pier can be calculated on the compressive strain of concrete.

Since the rotation angle of the joint section of the pier is extremely small in this status, the curvature $\varphi_y$ for the bottom section of the pier in the code [25] can be used for calculation.

$$\varphi_y = \frac{\varepsilon_c}{b} \approx \frac{1.975\varepsilon_y}{b} \tag{15}$$

Assuming that the curvature of the pier varies linearly with the pier height, with a curvature of 0 at the top section and $\varphi_y$ at the bottom section, the resulting displacement at the pier's top due to flexural deformation can be calculated as follows:

$$\Delta_e = \int_0^L x \cdot \varphi(x) dx = \frac{\varphi_y L^2}{3} \tag{16}$$

The total displacement of the top of the pier is calculated as:

$$\Delta_y = \theta_y L + \Delta_e \tag{17}$$

By taking the moment of reasonable force point for concrete, it is possible to obtain the bending moment at the bottom of the pier in the yield status.

$$M_y = W\left(\frac{b - c_y}{2} - \Delta_y\right) + \sum F_{PTi}\left(d_{PTi} - \frac{1}{3}c_y\right) + \sum F_{si}\left(d_{si} - \frac{1}{3}c_y\right) \tag{18}$$

The horizontal force at the top of the pier in the yield status is calculated as:

$$F_y = M_y L^{-1} \tag{19}$$

### 2.2.3. Ultimate Status

As the increases of lateral force acting on the pier top, the ED bars in tension would transfer from the yield status to the strengthening stage, resulting in higher tensile stress. The opening angle of the joint section and the curvature of the compressed concrete would be increased, leading to a continued increase in compression strain, as depicted in Figure 6. Hence, under lateral loads, the damage to the RSC bridge pier primarily occurs in the tensile ED bar and the compressed concrete in the compressed zone.

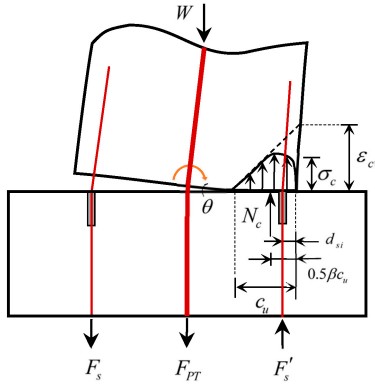

**Figure 6.** Bridge pier base in ultimate status.

The structural form of the RSC bridge pier, the reinforcement ratio of the ED bars, the unbonded length, the arrangement of steel bars, and other factors differ from each other for different piers, so the ultimate status of the RSC bridge pier needs to be analyzed based on

the specific situation of the bridge pier. The ultimate compression strain $\varepsilon_{cu}$ and ultimate tensile strain $\varepsilon s_u$ are the maximum strains that the concrete and reinforcement materials can withstand before failure occurs. The calculation process will stop once one of these ultimate statuses is reached. This decision is made in order to ensure the structural safety and stability of the pier and prevent any damage that could impact its performance [26–28].

The force $N_c$ acting on the compressed zone at the bottom of the pier by the abutment within the height range of $c_u$ can be calculated as:

$$N_c = \int_0^{c_u} h\sigma_c dx \tag{20}$$

For confined concrete, the model proposed by Mander [29] can be used. To simplify calculations, the complex stress distribution of the compressed concrete can be replaced by an equivalent rectangular stress distribution.

The total force of the rectangular stress distribution is $\alpha f_c \beta c_u h$, where $h$ is the height of the section perpendicular to the calculation direction. The actual height of the compressed zone is $c_u$, and $\alpha$ and $\beta$ are correlation coefficients of the equivalent rectangular stress distribution [24].

$$N_c = \alpha f_c \beta c_u h \tag{21}$$

According to Equation (8), the height of the compressed zone $c_u$ can be calculated as:

$$c_u = \frac{N_c}{\alpha f_c \beta b} \tag{22}$$

The opening angle $\theta_u$ at the joint can be calculated as:

$$\theta_u = \frac{\varepsilon_{su}(L_s + 2L_{eu})}{d_i - c_u} \tag{23}$$

To determine the working state of concrete, the monolithic beam analogy proposed by Pampanin [18] can be used to calculate the compressive strain of concrete at the edge of the pier column:

$$\varepsilon_c = \left( \frac{\theta L}{(L - \frac{L_p}{2})L_p} + \varphi_y \right) \times c \tag{24}$$

where $\varepsilon_c$ represents the compressive strain of the concrete at the edge of the compressed zone, $c$ represents the height of the compressed zone, $\theta$ represents the angle of the joint, $\varphi_y$ represents the yield curvature at the bottom of the pier, and $L_p$ represents the equivalent plastic hinge length.

The displacement at the top of the pier due to flexural deformation at this time can be calculated as:

$$\Delta_e = \frac{\varepsilon_c}{b} \times \frac{L^2}{3} \tag{25}$$

The displacement at the top of the pier in the ultimate status can be calculated as:

$$\Delta_u = \theta_u L + \Delta_e \tag{26}$$

The calculation of prestressed reinforcement and ED bar has been previously described in detail and will not be repeated here.

The total ultimate bending moment at the bottom of the pier can be calculated as:

$$M_u = W\left(\frac{b - c_y}{2} - \Delta_u\right) + \sum F_{PTi}\left(d_{PTi} - \frac{1}{2}\beta c_u\right) + \sum F_{si}\left(d_{si} - \frac{1}{2}\beta c_u\right) \tag{27}$$

The force at the top of the pier under ultimate status can be calculated as:

$$F_u = M_u L^{-1} \tag{28}$$

### 2.2.4. Double-Column Pier and Circular Pier

When analyzing a double-column pier, the presence of a capping beam with finite flexibility makes it more challenging to determine the displacement compared to a single-column bent. However, the position of the inflection point in a twin-column pier is only affected by the stiffness of the capping beam, and the impact of the distance between the columns can be disregarded, according to [30].

The stiffness of the capping beam affects the position of the point of inflection the column in pier. The pier is divided into two parts by calculating the position of the anti-bending point, and each part can be analyzed separately as a single-column pier and then superimposed. It is crucial to iteratively calculate the prestressing steel bars interspersed in the pier column to ensure that the prestressing force is the same in the upper and lower parts.

Figure 7 illustrates the effect of different capping beam stiffnesses on the inflection point of the columns. For example, a rigid beam, as shown in Figure 7a, can divide the column into two parts for individual calculations, whereas a flexible beam, as shown in Figure 7b, can be analyzed as a single-column pier.

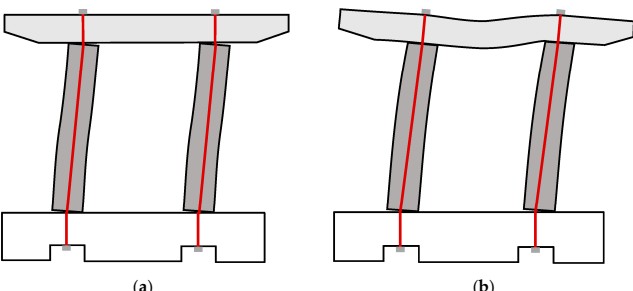

(**a**)                  (**b**)

**Figure 7.** Deformation mode of RSC double-column pier. (**a**) Rigid capping beam (**b**) flexible capping beam.

For circular bridge piers (Figure 8), the calculation method from the initial status to the decompression status is similar to that described above. When the angle of the joint section of the pier is open, the joint's cross section takes on the shape shown in Figure 9, where the shaded region represents the compressed zone.

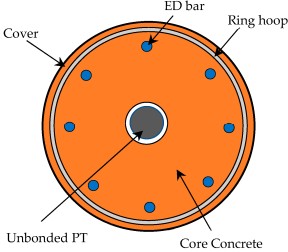

**Figure 8.** Cross section taken at circular pier joint.

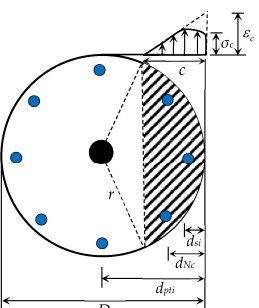

**Figure 9.** Compressed area at circular pier joint section.

The area of the compression zone in the circular pier section with a compressive height of $c$ can be calculated as:

$$A_c = \cos^{-1}(\frac{r-c}{r}r^2) - \sqrt{r^2 - (r-c)^2}(r-c) \tag{29}$$

The resulting force provided by the compressed concrete in the area is calculated as:

$$N_c = \int_0^c 2\sigma_c \sin\cos^{-1}(\frac{r-c}{r})dx \tag{30}$$

Based on the equivalent rectangular stress block theory, the equivalent stress is assumed to be 0.85 $f_c$. The compressive zone height $c$ at the bottom section of the pier is determined using the static equilibrium condition. The distance from the point of action of the resultant force to the compressed edge can be calculated as follows:

$$d_{N_c} = \frac{2\int_0^{\beta c} x\sqrt{r^2 - (r-x)^2}}{A_c}dx \tag{31}$$

After obtaining the distance between the point of application of the resultant force and the compressed edge at the bottom section of the pier, the bending moment at the bottom of the pier can be determined using a method similar to Equation (27). This allows for the calculation of the force–displacement relationship of the pier based on the bottom bending moment.

### 2.2.5. Flowchart of Calculation Procedure

Figure 10 shows the calculation flowchart of the above method.

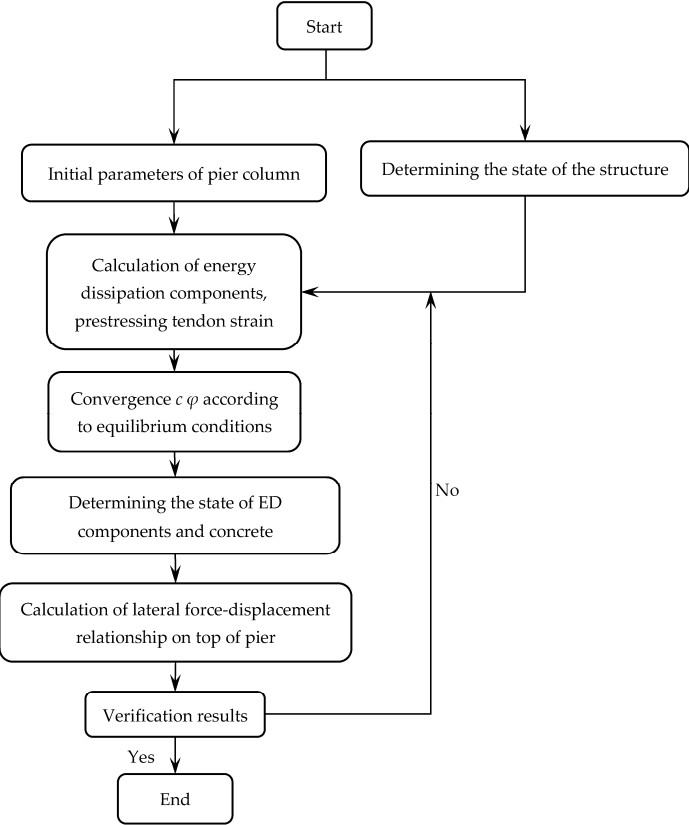

**Figure 10.** Flowchart of calculation procedure.

## 3. Verification of the Proposed Method

The skeleton curve is the backbone of the hysteresis loop and represents the carrying capacity of the system. In order to verify the accuracy of the calculation method, six single-column piers and two double-column piers were selected for comparative testing, which are shown in Figure 11a–h. The selected rocking bridge pier tests encompassed different types of structures and various energy dissipation components, fully demonstrating the applicability of the calculation method. Table 1 provides the basic information of the test specimens.

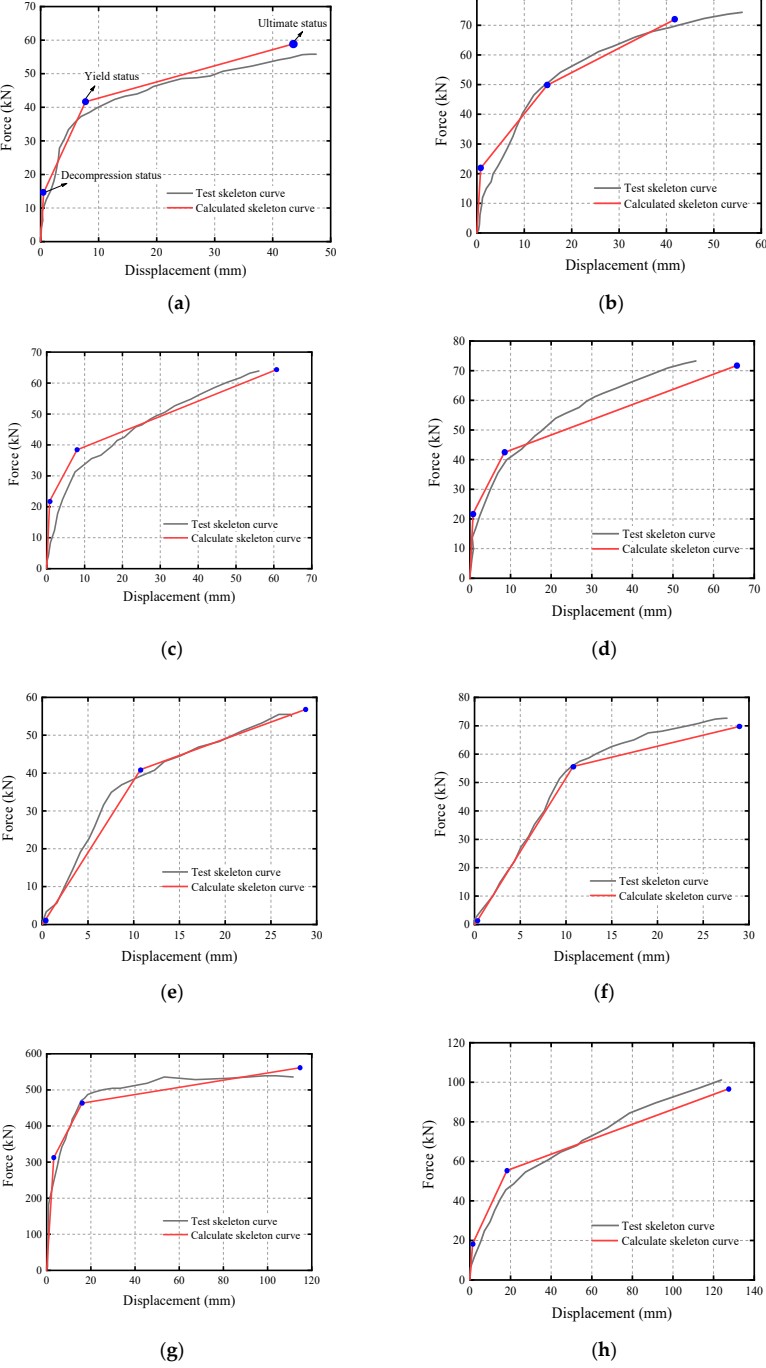

**Figure 11.** Comparison of the skeleton curves between calculated and test results. (**a**) Specimen HBD1, (**b**) specimen HBD2, (**c**) specimen HBD3, (**d**) specimen HBD4, (**e**) specimen G75W, (**f**) specimen G140W, (**g**) specimen TRB-B, (**h**) specimen FRP1.

**Table 1.** Basic information of test specimens.

| Test Specimens | Structure Form | ED Components | Ultimate Status |
|---|---|---|---|
| HBD1 | Single-column rectangular rocking pier | ED bar | Failure of ED components |
| HBD2 | Single-column rectangular rocking pier | ED bar | Failure of ED components |
| HBD3 | Single-column rectangular rocking pier | Anti-flexing mild steel | Concrete crush |
| HBD4 | Single-column rectangular rocking pier | Anti-flexing mild steel | Failure of ED components |
| G75W | Single-column with rectangular expansion base round rocking pier | Shape memory alloy (SMA) washer springs | SMA "locked" |
| G140W | Single-column with rectangular expansion base round rocking pier | Shape memory alloy (SMA) washer springs | SMA "locked" |
| TRB-B | Double-column rectangular rocking pier | Mild reinforcing bar | Failure of ED components |
| FRP1 | Double-column round rocking pier | None | Concrete crush |

Palermo [18] conducted static cyclic tests on rocking columns and studied the effect of different ratios of ED reinforcement and different initial tension on the rocking columns. Figure 11a,b show the comparison of the calculation results and the test skeleton curves of the HBD1 and HBD2 test specimens. Marriott [19] replaced the ED components on the basis of Palermo's research and conducted experimental studies. Figure 11c,d show the comparison of the calculation results and the test skeleton curves of the HBD3 and HBD4 test specimens.

Cheng [31] studied a new type of RSC bridge pier system based on memory alloy spring washers. Unlike conventional RSC bridge piers, this structure does not forcibly set unbounded prestressed reinforcement. Its memory alloy spring washers provide sufficient self-centering and ED capabilities required for rocking. Figure 11e,f show the comparison of the calculation results and the test skeleton curves of the G75W and G140W test specimens.

Han Qiang [32] conducted experimental studies on a scale model of the Huangxu Road RSC bridge pier. The results of the calculation method in this paper are consistent with the test results of the TRB-B double-column test specimen, as shown in Figure 11g. Eigawady [7] conducted tests on rocking double-column circular piers, and the results of the calculation method in this paper have also been verified by the test results of the rocking double-column circular piers, as shown in Figure 11h. Due to the lack of ED components in the specimen, we defined the yield status of the specimen as the point where the compressive edge concrete reaches peak stress.

The above method involves determining and calculating three statuses, connected by straight lines, as shown in the comparison chart of skeleton curves, which displays a clear three-segmented line. The first inflection point signifies the decompression status, the second inflection point represents the yield status, and the final point denotes the ultimate status, as depicted in Figure 11. The results obtained from this calculation method exhibit good agreement with the skeleton curves and effectively reflect the behavior of the experimental skeleton curves.

Table 2 displays the experimental and calculated data for the three status of each test specimen. The yield point of the test skeleton curve was determined using the farthest-point method [33]. The ultimate status was determined by using the peak load point of the experimental skeleton curve, as reported in the literature.

When it comes to determining the "decompression status," it often appears in the early stage when the structure is usually in the elastic phase, making it difficult to clearly identify the location of the decompression status on the skeleton curve. In order to pin-point its location, the farthest-point method can be utilized once again to find the right point between the starting point and yield point, and accordingly determine the decom-pression status on the skeleton curve. Due to the difficulty in selecting the decompression status, there may be a large difference between the calculated value and the experimental value.

**Table 2.** Experimental and calculation data.

| Status | Force Displacement | Value | HBD1 | HBD2 | HBD3 | HBD4 | G75W | G140W | TRB-B | FRP1 |
|---|---|---|---|---|---|---|---|---|---|---|
| Decompression | Displacement (mm) | Test | 0.67 | 3.34 | 3.16 | 0.93 | 0.21 | 0.12 | 1.72 | 1.97 |
| | | Calculation | 0.53 | 0.78 | 0.67 | 1.04 | 0.07 | 0.13 | 3.24 | 1.44 |
| | | Test/ Calculation | 1.26 | 4.28 | 4.72 | 0.89 | 3.00 | 0.92 | 0.53 | 1.37 |
| | Force (kN) | Test | 10.46 | 19.88 | 19.59 | 15.48 | 1.32 | 1.89 | 196.77 | 11.68 |
| | | Calculation | 14.58 | 22.08 | 21.53 | 22.41 | 0.23 | 0.42 | 312.27 | 18.21 |
| | | Test/ Calculation | 0.72 | 0.90 | 0.91 | 0.69 | 5.74 | 4.50 | 0.63 | 0.64 |
| Yield | Displacement (mm) | Test | 8.41 | 17.61 | 7.49 | 15.99 | 8.68 | 9.93 | 21.65 | 27.32 |
| | | Calculation | 7.73 | 14.83 | 8.01 | 8.48 | 10.73 | 10.77 | 16.04 | 18.27 |
| | | Test/ Calculation | 0.92 | 0.84 | 1.07 | 0.53 | 1.24 | 1.08 | 0.74 | 0.67 |
| | Force (kN) | Test | 38.45 | 54.16 | 31.26 | 47.97 | 36.94 | 53.77 | 494.25 | 54.55 |
| | | Calculation | 41.65 | 49.81 | 38.43 | 42.45 | 40.89 | 55.65 | 463.38 | 55.39 |
| | | Test/ Calculation | 1.08 | 0.92 | 1.23 | 0.88 | 1.11 | 1.03 | 0.94 | 1.02 |
| Ultimate | Displacement (mm) | Test | 47.48 | 54.87 | 56.07 | 55.64 | 25.86 | 27.32 | 111.60 | 123.88 |
| | | Calculation | 43.50 | 41.58 | 60.57 | 65.53 | 28.69 | 28.92 | 114.34 | 127.08 |
| | | Test/ Calculation | 0.92 | 0.76 | 1.08 | 1.18 | 1.11 | 1.06 | 1.02 | 1.03 |
| | Force (kN) | Test | 56.69 | 78.11 | 63.91 | 73.27 | 55.49 | 72.66 | 535.64 | 101.18 |
| | | Calculation | 58.82 | 71.83 | 64.25 | 71.64 | 56.74 | 69.71 | 561.04 | 96.58 |
| | | Test/ Calculation | 1.04 | 0.92 | 0.99 | 0.98 | 1.02 | 0.96 | 1.05 | 0.95 |

## 4. Conclusions

(1) This article simplifies the force–deformation response of the RSC bridge piers into three stages by analyzing the whole load–deformation condition and making reasonable assumptions. The proposed method presents an analysis and calculation procedure for the RSC bridge piers in decompression, yielding, and ultimate status. Compared to other methods, such as the equivalent plastic hinge model, the proposed method is more consistent with the actual load–deformation response of the RSC bridge piers and is simpler to calculate, thus reducing the application difficulty in engineering. However, some limiting factors may affect its accuracy when providing guidance for engineering practice, such as material properties, manufacturing defects, forms of seismic loading, etc.

(2) The accuracy of the method is verified by comparing the calculated force–displacement relationship of the RSC bridge piers with the tested skeleton curves of six single-column piers and two double-column piers. The proposed method can serve as an easy and powerful tool to evaluate loading–deformation curves that have good agreement with the experiments. The proposed formula for force–displacement relationship in this article reveals the force mechanism and characteristics of structural rocking behavior, providing a calculation idea for seismic design of the RSC bridge and a theoretical basis for the engineering application of RSC bridge piers.

(3) The number of specimens used to validate the method is relatively limited in this article. More specimens may be required to more comprehensively validate the proposed method in future research, as well as to further discuss the hysteresis rules associated with this method. Although researchers have studied the seismic behavior of RSC piers, further theoretical research and analysis are needed to fully understand its mechanical properties and performance characteristics, especially for complex structures (such as double-column piers, multicolumn piers, etc.) and complex load effects. Moreover, most of the existing RSC bridge piers are based on theoretical analysis and small-sample tests, lacking long-term performance research and actual measurement data, particularly in terms of durability and fatigue. Subsequent research should be focused on these shortcomings.

**Author Contributions:** Conceptualization, S.W. and Z.S.; methodology, S.W. and Z.S.; validation, S.W., Z.S. and D.W.; formal analysis, S.W., Z.S. and D.W.; writing—original draft preparation, S.W., Z.S. and D.W.; writing—review and editing, S.W. and Z.S.; supervision, D.W.; funding acquisition, Z.S. All authors have read and agreed to the published version of the manuscript.

**Funding:** This study was supported by the Key Research and Development Program of Hebei Province of China (grant 21375405D), supported by Fundamental Research Funds for the Central Universities of China (grant ZY20215111), the National Natural Science Foundation of China (grant 51978167), and the Science and Technology Innovation Program for Postgraduate Students in IDP subsidized by Fundamental Research Funds for the Central Universities (grant ZY20230313).

**Institutional Review Board Statement:** Not applicable.

**Informed Consent Statement:** Not applicable.

**Data Availability Statement:** Not applicable.

**Conflicts of Interest:** The authors declare that they have no known competing financial interests or personal relationships that could have appeared to influence the work reported in this paper.

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
