# Peer review of "Analysis and Verification of Load–Deformation Response for Rocking Self-Centering Bridge Piers"

_sustainability, doi:10.3390/su15108257_

Round 1

Reviewer 1 Report

This article simplifies the force process of RSC bridge piers into three stages by analyzing the entire force process and making reasonable assumptions. The proposed method presents an analysis and calculation process for RSC bridge piers in the decompression status, yielding state, and design limit state. Compared to previous calculation methods, such as the equivalent plastic hinge model, this method is more consistent with the actual deformation behavior of RSC bridge piers and is simpler to calculate, thus reducing the application difficulty in engineering.

Section 1. The research gap should be clearly stated. What methods are people in the industry using? What theoretical approaches have been developed but not widely used?

Line 118. "Where" -> "where" and delete the space before it (because this is NOT a new paragraph).

"Furthermore, the design theory and calculation methods of RSC bridge piers are not yet mature and require further research and exploration." -> "Furthermore, the design theory and calculation methods of RSC bridge piers are not yet mature and require further research and exploration. Integrated machine learning and optimization approaches are viable for this direction (Yan et al., 2021; Tian et al, 2023)"

Yan, R., Wang, S., Cao, J., Sun, D., 2021. Shipping domain knowledge informed prediction and optimization in port state control. Transportation Research Part B 149, 52–78.

Tian, X., Yan, R., Wang, S., Liu, Y., Zhen, L., 2023. Tutorial on prescriptive analytics for logistics: What to predict and how to predict. Electronic Research Archive 31(4), 2265–2285.

Line 131. Delete the space at the beginning. 

Please carefully proofread your paper. For example, you have:

2. Analysis of the full process of force on rocking self-centering bridge piers

2 Verification of the test

I will have to recommend rejection if there are such careless mistakes in the next round.

18 figures are too many. Combine some of them. 

None.

Reviewer 2 Report

Dear authors,

The paper is interesting but needs to be adjusted in some places before publication and below are some suggestions:
- Key words to be put in alphabetical order
- Line 113 and 114 can not be noted with 1.2....above there was 2.1....more that from an editorial point of view it does not look good on the last lines of the page to be the subtitle
- Same with line 142, perhaps a renumbering would be good
- Line 195 the heading "Ultimate Status" might need to be changed a bit
- Subchapter 2 also appears in line 57 and line 269
- Perhaps it would be better to divide chapter 2 of the paper into two subchapters, one on "Methods applied" and the second on "Analyses".
- The conclusions part would have to be extended
- Check if all references are included in the text in chronological order and it would be recommended that the references be as many as possible from the last 5 years.

 Moderate editing of English language

Reviewer 3 Report

This article presents a useful method for predicting the force-displacement curve of RSC bridge piers. The predicted results by the method show good consistency with the testing ones, demonstrating the accuracy and practicality of the method. This research could present valuable insights for developing simplified analysis methods for RSC bridge piers. Although the article is well-written and interesting, but some revisions should be conducted to make it better. Here are my suggestions:

1.The title should be more clear and concise. I suggest changing “Full process of force acting” to “Load-deformation response”.

2.The abstract should be treated seriously, as it can attract the attention of readers. There are some errors in tenses in the abstract, such as using “was” instead of “is”. Please check the rest of the article carefully for grammar and spelling errors. In line 14, some words can be replaced to make the sentences more accurate, such as using “actions” instead of “condition”.

3.The logic of section 2.1 is slightly confusing. the description of the rotational deformation behavior of the RSC piers is unclear and the statements are imprecise. Please sort it out and focus on introducing the mechanical mechanism of RSC bridge piers.

4.The paper contains many grammatical errors. For example, in line 90, "fundamental contrast" should be revised to "main differences" to accurately depict the difference between RSC piers and ordinary piers. In line 92, "response" should be substituted for "motion"; and "relaxing" is more suitable than "motion". Additionally, in line 92, "relaxing" should be employed rather than "loosening" to enhance the semantic expression. It is recommended that the entire text be carefully reviewed for errors. The English writing quality requires significant improvement.

5.The proposed three-stage calculation method for analyzing the force-deformation response of RSC bridge piers is a critical aspect of this paper, and as such it is recommended that the authors provide a detailed and accurate discussion of the method in the text. Specifically, the authors are encouraged to provide a thorough explanation of the formulas used, and the meaning and algorithms of each parameter so that readers can easily reproduce the calculations following the detailed steps provided in the text.

6.In Figures 11-18, why are the values of Decompression status in Figure 15 and Figure 16 so small compared to other figures? Please explain. Please mark each turning point clearly in the figures. Also, please renumber these figures according to the journal requirements.

7.The third part of Conclusions, “This lack of data is not conducive to the reliability and safety evaluation of structures, and the comprehensive verification of its application effect has not been carried out in practical engineering. Furthermore, the design theory and calculation methods of RSC bridge piers are not yet mature and require further research and exploration.” is not a research conclusion of this paper. I suggest moving it to a separate section or deleting it.

Reviewer 4 Report

The article provides a clear and comprehensive analysis of the behavior of RSC bridge piers. Its significant contribution lies in its approach to analyzing the force-displacement response, considering both rotational and bending deformation. It simplifies the force loading process into three stages and explains the calculation method for top lateral force and displacement of single-column and double-column RSC bridge piers. The proposed method is validated by comparing calculated values with experimental data, and it is both interesting and easy to use for engineers. Here are some suggestions:

1.In the Introduction, it is suggested to further add references that are relevant to this paper, so that readers can better understand the research progress in this direction. To facilitate readers’ search for papers, it is suggested to enrich the keywords as much as possible.

2.It is noted that your manuscript needs careful editing by someone with expertise in technical English editing paying particular attention to English grammar, spelling, and sentence structure so that the goals and results of the study are clear to the reader.

3.There are many characters and data in the text, please carefully proofread each symbol, including capitalization, italicization, subscripts and superscripts, pay attention to the distinction between subscripts and lowercase letters, etc., and ensure the accuracy of the characters.

4.Please refer to the journal requirements for figures and formulas in the text. Figures should have clear and identifiable lines and data.

5.In the line 192, changing “process” to “procedure” would be more appropriate.

6.By observing Figures 11-18, the data of “Decompression status” in Table 2 corresponds to the data of “Yield status”. Please supplement the data of “Decompression status” in Table 2. Complete and accurate data is meaningful for the dissemination of the article and the reading of readers.

7.Figures 11-18 show a comparison of the Skeleton Curves between the Calculated and Test Results, and they have the same function. It is recommended to renumber these figures according to the writing format of this journal. For a concise explanation of the comparison in the diagrams, the name of each specimen can be specified after the word "Specimen" in the provided diagrams.

8.Figure 7(a) and Figure 7(b) are not easily understood, thus necessitating a redrawing.

9.It should be explained how to determine the yield state of FRP1 since the test specimen does not have ED components. 

10.Furthermore, in the second section of the conclusion, since the Displacement-based aseismic design method is not examined in this study, it is recommended that "based on displacement" be removed from the conclusion.

Overall, I think this is a valuable paper that deserves publication after minor revisions.

Reviewer 5 Report

In this paper, a three-stage analysis method is proposed to calculate the lateral force-displacement relationship of the steel embedded RSC rectangular bridge pier. However, the reviewers raised doubts about the accuracy of the method in comparison with experimental results. Followings are the comments and questions for the author’s consideration:

1The Introduction introduces the importance of seismic performance of bridges, compares the seismic design theories of conventional reinforced concrete piers and RSC piers, and lists some previous works, suggesting the characteristics and limitations of previous studies in detail.
2
The combined force of the ED reinforcement under pressure at the joints is ignored in the calculation. Is there any data to support this?

3The force-displacement relationships of the bridge piers calculated using the method were then compared to the experimental skeleton curves, which are shown in Figures 11 to 18. The reason for the large error should be explained.

4There are repeated syntax errors and formatting errors in statements, such as

1) Line18 The word 'displacement' is repeated.

2) Line28 The word 'which' should be preceded by a comma.

3) Line44-48 Sentence ‘However, due to the "opening-closing" process of RSC piers during earthquake action, the formation of a plastic hinge at the bottom of the pier may be difficult.’ and sentence ‘However, it may be difficult to form a plastic hinge at the bottom of the pier due to the "opening-closing" process of RSC piers during earthquake action.’ are semantically duplicated.

4) Line60-67 Sentence ‘Which assumes that the compressed zone height is zero. While this method offers a simple model and calculation process, it significantly overestimates the initial stiffness and  strength of RSC bridge piers, leading to design results that are biased towards being unsafe.’ is repeated 2 times.

5) Line68 Wrong position of reference symbol [11]

6) Line100-105 Sentence ‘’ is repeated 2 times.

7) Confusing subheading serial numbers

 Please check through the typos and minor editing of English language is required.

Round 2

Reviewer 2 Report

 Accept in present form

 Accept in present form